# Low-Dose Tacrolimus Promotes the Migration and Invasion and Nitric Oxide Production in the Human-Derived First Trimester Extravillous Trophoblast Cells In Vitro

**DOI:** 10.3390/ijms23158426

**Published:** 2022-07-29

**Authors:** Ahmad J. H. Albaghdadi, Kassandra Coyle, Frederick W. K. Kan

**Affiliations:** Department of Biomedical and Molecular Sciences, Faculty of Health Sciences, Queen’s University, Kingston, ON K7L 3N6, Canada; a.albaghdadi@queensu.ca (A.J.H.A.); 15kmc11@queensu.ca (K.C.)

**Keywords:** tacrolimus, extravillous trophoblasts, HTR-8/SVneo cells, nitric oxide, nitric oxide synthase, progesterone receptors, recurrent pregnancy loss, uterine spiral artery remodeling

## Abstract

Placentation is one of the most important determinants for a successful pregnancy, and this is dependent on the process of trophoblast migration and invasion. Progesterone receptors (PGR) are critical effectors of progesterone (P4) signaling that is required for trophoblast migration and invasion conducive to a successful gestation. In immune complicated pregnancies, evidence has shown that abnormal placentation occurs because of aberrant expression of PGR. Therapeutic intervention with tacrolimus (FK506) was able to restore PGR expression and improve pregnancy outcomes in immune-complicated gestations; however, the exact mode of action of tacrolimus in assisting placentation is not clear. Here, we attempt to uncover the mode of action of tacrolimus by examining its effects on trophoblast invasion and migration in the human-derived extravillous trophoblast (EVT) cell line, the HTR-8/SVneo cells. Using a variety of functional assays, we demonstrated that low-dose tacrolimus (10 ng/mL) was sufficient to significantly (*p* < 0.001) stimulate the migration and invasion of the HTR-8/SVneo cells, inducing their cytosolic/nuclear progesterone receptor expression and activation, and modulating their Nitric Oxide (NO) production. Moreover, tacrolimus abrogated the suppressive effect of the NOS inhibitor N^ω^- Nitro-L-Arginine Methyl Ester (L-NAME) on these vital processes critically involved in the establishment of human pregnancy. Collectively, our data suggest an immune-independent mode of action of tacrolimus in positively influencing placentation in complicated gestations, at least in part, through promoting the migration and invasion of the first trimester extravillous trophoblast cells by modulating their NO production and activating their cytosolic/nuclear progesterone-receptors. To our knowledge, this is the first report to show that the mode of action of tacrolimus as a monotherapy for implantation failure is plausibly PGR-dependent.

## 1. Introduction

In mammalian reproduction, placentation is one of the key determinants of a successful pregnancy. The foundation of this process is dependent on trophoblast migration and invasion as the extent of the invasion ultimately determines the quality of the anchorage [1]. This, in turn, controls the pregnancy outcome. Post-implantation invasion of the trophoblasts is a multi-step process that first involves the attachment of trophoblast cells to the extracellular matrix (ECM) [1]. This is followed by invasion of the decidual and degradation of the endothelial ECM, and finally migration of the cells through the eroded tissue [2]. One component of this process is the trophoblast invasion of the maternal spiral arteries, which is accomplished by the subsequent remodeling of the spiral artery wall ensuring a secure maternofetal connection, and an adequate supply of oxygen and nutrients to the implanted fetus [3]. The entire process of trophoblast invasion is tightly controlled under the influence of maternal hormones, with progesterone playing a crucial role among others [2]. Dysfunction of this control process, manifested as either insufficient or excessive invasion, could further lead to several gestational complications including miscarriage, pre-eclampsia, fetal growth restriction (FGR), placental abruption and intrauterine death [1,3].

Progesterone (P_4_) is the principal steroid hormone of pregnancy and exerts genomic actions in the female reproductive tract to produce tissue- and cell-specific responses [4]. In hemochorial species, P_4_ primarily binds to its cognate receptor, the progesterone receptor (PGR), to exert a myriad of transcriptional activities [5]. In the absence of a ligand, steroid receptors are not confined to a particular cell compartment and are free to shuttle continuously between the cytoplasm and nucleus [6]. As a class I nuclear receptor, PGR primarily resides in the cytosol in the absence of a ligand. Therefore, for transcription to be initiated, PGR must first be translocated into the cell nucleus via binding to its progesterone ligand. Following translocation, the genomic actions of progesterone are then mediated via two isoforms of nuclear PGRs: PGR-A (94kDa) and PGR-B (116kDa) [7]. These isoforms are differentially expressed in hemochorial species, with PGR-A mainly being localized in the endometrium and PGR-B being dominant in the myometrium [5]. Although PGR-A and PGR-B have similar steroid hormone binding activities, the transcriptional activities that they regulate are distinct [7]. For instance, PGR-A generally induce pro-inflammatory responses through NFkB-regulated genes and IL1B, whereas PGR-B exerts anti-inflammatory responses [5]. Moreover, in vitro experiments have shown that when PGR-A and PGR-B proteins are co-expressed in cultured cells, PGR-A can impede with the signaling activities of PGR-B and vice versa [7]. Previous research has shown that cellular responses to progesterone may be modified by the ratio of the expression of each PGR isoform in the body [7]. Since both PGR-A and PGR-B are involved in the fine tuning of molecular events leading to a successful implantation, the regulation of these two isoforms is vital.

Nitric oxide (NO) is a small molecular weight mediator with diverse functions, including vasodilation, inhibition of platelet aggregation and promotion of vascular remodeling [8]. Nitric oxide synthase (NOS) is expressed on the EVT [8,9,10,11], and is the enzyme involved in the formation of NO from the conversion of L-arginine to L-citrulline [12]. Local production of NO by the EVT cells is believed to play a crucial role in placental vascular development as it aids in uterine spiral arterial remodeling (SAR), promotes migration and invasion of EVTs and contributes to vasodilation within the placental vasculature [10,13,14,15]. The formation of NO from eNOS is dependent on Ca^2+^/calmodulin interactions and is regulated by numerous factors including changes in phosphorylation status [16]. Phosphorylation is a major posttranslational modification, and the regulation of eNOS by phosphorylation is a highly complex process. Seven regulatory phosphorylation sites influencing NOS activity have been recognized in bovine eNOS at Y83, S116, T497, S617, S635, Y659, and S1179 with equivalent, functional sites also found in human eNOS at Y81, S114, T495, S615, S633, Y657, and S1177 [16,17,18]. One possible mechanism for the phosphorylation of eNOS involves Serine 1177-1179 (Ser1177-1179), which represents a major positive regulatory domain of eNOS [18]. Phosphorylation of Ser1177-1179 is catalyzed by several distinct kinases including AkT, PK-A, AMPK, or the Ca^2+^/calmodulin-binding domain and has been shown to increase NO production [16,18]. This underpins Ser1177-1179 phosphorylation as highly integrative and central posttranslational modification as the different individual kinases themselves can in turn be regulated by very diverse stimuli [16]. Another phosphorylation site regulating the enzymatic activity of eNOS involves Threonine 495 (Thr495) [16]. Thr495 is located in the Ca^2+^/calmodulin-binding domain, representing the major negative regulatory site of eNOS [16]. Phosphorylation of Thr495 attenuates eNOS activity by interfering with calmodulin binding due to repulsive steric or charge effects [16]. The phosphorylation of eNOS can also occur through tyrosine-mediated events. However, this mechanism is unresolved as to whether tyrosine phosphorylation is involved in modulating enzyme activity or in providing binding sites for proteins with Src homology or a phospho-tyrosine binding domain [16].

We have previously shown that the use of a low-dose (0.1 mg/kg) tacrolimus (formerly known as FK506; a 822 kDa naturally occurring lipophilic macrolide lactone immunosuppressant that was first isolated from the soil bacterium *Streptomyces tsukabaensis* [19] restores placentation and prevents uterine SAR defects in a murine model of FGR and recurrent implantation failure [20]. Recently, tacrolimus has also been used to treat female infertility and prevent adverse pregnancy outcomes in women with recurrent pregnancy loss with elevated systemic Th1 (CD4+ IFNγ+): Th2 (CD4+IL4+) cell ratios [21,22,23]. The well-characterized mode of action of tacrolimus is the inhibition Ca^++^ dependent activation of NFκB and the nuclear factor of activated T cells (NFAT) in T lymphocytes blocking T-cell receptor-mediated lymphokine gene transcription, degranulation, exocytosis and apoptosis [24,25]. Therefore, tacrolimus is more commonly used clinically to reduce the risk of rejection in recipients of allogeneic organ transplants [19]. However, tacrolimus has been found to have another proposed mode of action. As a FK506 binding protein, tacrolimus is capable of binding other members of the FKBP family identified in cells to affect multiple cellular and molecular pathways [26]. The tetratricopeptide repeats (TPR)-containing FKBP52 present in steroid chaperone complexes are well-known intracellular targets for tacrolimus [27,28]. Besides serving as a target, FKBP52 is also a cochaperone for the PGR complex that functions to govern and potentiate its hormonal actions in target tissues [29]. For PGR to function appropriately, it has to be able to maintain a functional state that is competent for hormone binding and nuclear shuffling which are dependent on its interactions with molecular chaperone machinery [4,30,31,32]. One model outlining tacrolimus’ mode of action proposes that the binding of tacrolimus facilitates the release of the receptor from the chaperone complex [30]. In doing so, this allows PGR to be shuttled into the intranuclear space to act, thereby increasing the bioavailability of PGR for downstream signaling with P4 and promoting PGR signaling activity [30]. In a murine model of FGR, we previously described abnormalities in the expression of the PGR and FKBP52 to be one of the major causes of implantation failure, improper uterine SAR and FGR [5,20]. Moreover, we have also shown that the tacrolimus-mediated restoration of the PGR pathway in these complicated pregnancies is contingent upon the induction of the expression and proper cellular localization of the PGR and the PGR-FKBP52 complex during implantation and placentation [5,20].

To date, there are no data available on the potential regulatory role of tacrolimus on the PGR signaling and NO synthesis in the human-derived EVT cells. Previous pre-clinical and molecular studies provided conflicting evidence regarding the modulatory effects of tacrolimus on vascular NOS expression and activity [33,34,35]. Therefore, to further elucidate the mechanism of action of tacrolimus in promoting placentation and uterine SAR, this study examined the effects of low-dose tacrolimus (10 ng/mL) on certain PGR-mediated molecular pathways involved in trophoblast migration and invasion, as well as NO production in vitro. Specifically, the effects of low-dose tacrolimus on the migration and invasion of the human-derived first-trimester EVT cells (the HTR-8/SVneo), their release of NO and their expression of the eNOS, p-eNOS-Ser^1179^ and p-eNOS-Thr^495^ as well as their naïve and activated forms of the PGR, FKBP52 and STAT3 were investigated in the presence or absence of an NOS inhibitor, the N^ω^- Nitro-L-Arginine Methyl Ester (L-NAME). We have unveiled a potential protective effect of low-dose tacrolimus in rescuing the migration and invasion, as well as the expression and activation of eNOS and NO release by the human-derived first-trimester EVT cells cultured under nitrosative stresses in the presence of L-NAME. Moreover, we have provided molecular cues demonstrating the positive influential effect of tacrolimus in potentiating the PGR signaling in the human-derived HTR-8/SVneo cells in conditions mimicry of fetal growth restrictions in vitro. Our data support the beneficial therapeutic use of low-dose tacrolimus in preventing early gestational failure through promoting the migration and invasion of the human first-trimester EVT cells in susceptible pregnancies, thereof.

## 2. Materials and Methods

### 2.1. Experimental Design and Test-Article Formulations

This study involved experimentation with the immortalized human-derived first trimester extravillous trophoblast cell line HTR-8/SVneo cells generously provided by Dr. Charles Graham of Queen’s University, Kingston, Ontario, Canada. Prior to treatment, cells were cultured following established protocols [36] and harvested when they reached at least 80% confluence. All cell cultures (passages 77–81) used in this study were screened and determined to be mycoplasma-free using a MycoFluor^TM^ Mycoplasma Detection Kit (Cat: M7006, Life Technologies, Montreal, QC, Canada) according to the provider’s instructions. Besides their untreated control cultures, cells were cultured under 3 different experimental conditions: low dose (10 ng/mL) tacrolimus (B415260, Toronto Research Chemicals, Toronto, ON, Canada), L-NAME (50 mM/L) (N5751, Sigma-Aldrich, Oakville, ON, Canada) [37], or low dose tacrolimus and L-NAME in combination treatment. Both tacrolimus and L-NAME were dissolved in DMSO under sterile cell conditions according to the manufacturers’ instructions. The dosage of 10 ng/mL of tacrolimus was chosen because previous data have shown that this is the lowest and safest concentration that can be used [38]. L-NAME was used in the present study as a nitric oxide synthase inhibitor because it has been successfully shown to induce pre-eclampsia (PE)-like conditions in vitro [39,40]. Following treatment, cells under the three separate conditions were collected, respectively, after 12, 24 and 48 h for downstream protein expression analysis and the detection of NO production.

To establish whether the mode of action of tacrolimus in promoting the migration and invasion of the human-derived first trimester EVT cells is at least in part PGR-dependent, HTR-8/SVneo cells were also treated with Mifepristone (50 nM/mL dissolved in ethanol) (M8046, Sigma-Aldrich, Oakville, ON, Canada) for 12–48 h in the presence or absence of tacrolimus and were subjected to further analyses by WB and IF detections of PGR and FKBP52 and their cytosolic and nuclear co-localization. 

### 2.2. IncuCyte Scratch Wound Assay and In Vitro Live-Cell Migration Analysis

Migration of the HTR-8/SVneo cells was assessed in real-time using the IncuCyte^®^ live cell-migration-based scratch assay according to established protocols [36,41]. HTR-8/SVneo cells were cultured in RPMI-1640 medium supplemented with 5% FBS in 6-well culture plates (Sigma-Aldrich, Mississauga, ON, Canada) until confluence according to the manufacturer’s instructions. Cells were plated (1 × 10^5^ cells per well) onto 24-well plate (ImageLock Microplate, Essen Bioscience Inc., Ann Arbor, MI, USA) in RPMI-1640 medium and allowed to settle overnight at 37 °C and 5% CO_2_. A scratch was made on confluent monolayers using a 96-pin WoundMaker™ (BioScience Inc, Ann Arbor, MI, USA) and plates were washed in unconditioned RPMI-1640 to remove detached cells. Conditioned medium that contained, respectively, tacrolimus (10 ng/mL), L-NAME (50 mM/mL), and a combination of both was then added and plates were incubated at 37 °C and 5% CO_2_ in special scanning chambers. Each experiment was performed in triplicate and was repeated four times. Pseudo-colored and phase-contrast images were produced to identify the position of the wounded (cell-free) and unwounded (cell-occupied) zones and were automatically scanned and acquired by the IncuCyte™ Zoom software system every 2 h for 48 h. Label-free analysis of collected wound image data was carried out using IncuCyte^®^ Zoom Scratch Wound Analysis Software Application Module and data are presented as the Relative Wound Density (RWD, Eizen, v1.0 algorithm). RWD represents the spatial cell density in the wound area relative to the spatial cell density outside of the wound area at each time point (time-curve) and implies the speed of cells occupying the scratch wound area. The rate of wound closure was compared using the half-maximal stimulatory time (ST_50_) and area under the time course curve (AUC).

### 2.3. In Vitro Matrigel Invasion Assay

To determine the influence of low-dose tacrolimus (10 ng/mL) on the invasiveness of HTR-8/SVneo cells with and without the eNOS inhibitor L-NAME, or mifepristone, we modified a previously described in vitro trophoblast cell invasion assay [42] and used reconstituted reduced growth factor extracellular matrix (Matrigel) (GelTrexT^M^, Cat. # A1569601, Life Technologies, Montreal, QC, Canada) as the substrate for invasion. 100 μL of liquid Matrigel was applied to 8 μm-pore 24-well plate Polyethylene Terephthalate (PET) inserts (Falcon^TM^ Cell culture Insert, Fisher Scientific, Cat. # 08-771-21) and allowed to gel at 37 °C for 4 h in a tissue incubator under sterile conditions. Successively, untethered liquid and material were aspirated, and 5 × 10^4^ cells were placed on top of the solidified Matrigel within the insert. A brief 2-h exposure of tacrolimus with and without L-NAME or mifepristone was performed prior to plating the cells onto the inserts, and prolonged treatment was continued throughout the duration of the assay. Each experiment was performed in triplicate and was repeated two times. After 24 h of incubation, inserts were washed in PBS and Matrigel was removed using a wet cotton swab. Invaded cells within the filter were fixed for 10 min in Methacarn solution that contained 3 parts of methanol and 1 part of acetic acid (*v/v)* washed and stained for 10 min in 1% (*w/v*) H&E and washed again. Filters of the inserts were cut out and mounted onto a slide. The Invasion Index representing the relative invasiveness of the HTR-8/SVneo cells was determined after censoring the number of cells that invaded through the Matrigel after a 24-h incubation in the presence or absence of tacrolimus and L-NAME or mifepristone using DNR BioImaging Systems MF-ChemiBIS 3.2 and the highly automated and flexible image analysis software Totallab TL100.

### 2.4. Western Blot Analysis

To discern the mode of action of tacrolimus in promoting the migration and invasion of the HTR-8/SVneo cells and modulating their NO production, Western blotting was performed. Protein concentrations were determined by use of Bradford protein assay followed by Western blot analysis to detect the protein expression of eNOS (33-4600, ThermoFisher Scientific, Burlington, ON, Canada), p-eNOS-Ser1179 (36-9100, ThermoFisher Scientific, Burlington, ON, Canada), p-eNOS-Thr495 (SC-136519, Santa Cruz Biotech. Inc., Dallas, TX, USA), PGR (SC-166169, Santa Cruz Biotech. Inc, Dallas, TX, USA),), phospho-PGR (pPGR) (Ser294, MA1-414, Invitrogen, Burlington, ON, Canada), FKBP52 (AF4095, R&D Systems, Minneapolis, MN, USA), STAT3 (SC-8019, Santa Cruz Biotech.) and pSTAT3 (9145, Cell Signaling) in the HTR-8/SVneo cells under the different treatment conditions (see Appendix A for a list of antibodies used in the present study). In brief, cells were separated into aliquots containing 50 μg of protein and mixed with 5 μL of 6× SDS loading buffer. Samples were run on 7.5% polyacrylamide gels, and blots were transferred onto PVD membranes. Membranes were blotted with 5% non-fat milk, and subsequently probed with the appropriate dilutions of the primary and HRP-conjugated secondary antibodies for exposure and analysis of band intensities using the SuperSignal™ West Atto Ultimate Sensitivity Substrate (A38555, ThermoFisher Scientific, Ottawa, ON, Canada) and the ChemiDoc™ Imaging System. The loading control α-tubulin was used as a comparative control for all Western blot analyses done in this study.

### 2.5. Detection of NO Production in the HTR-8/SVneo Cells

Flow cytometry analysis was used to determine the effects of tacrolimus on NO production in HTR-8/SVneo cells. Confluent cells were treated for 24 h with tacrolimus, L-NAME and a combination of both, respectively, in serum-free RPMI-1640 medium before undergoing subsequent treatment with the NO sensitive fluorescent probe, 4-amino-5-methylamino-2′,7′-difluorofluorescein (DAF-FM) diacetate (10 μM final concentration in serum-free RPMI-1640 medium) (D23842, Invitrogen, Ottawa, ON, Canada) according to the manufacturer’s instructions. DAF-FM was used as it has been shown to greatly enhance the sensitivity for NO detection [43]. The interaction of DAF-FM and NO creates fluorescent benzotriazole compounds that become trapped in the cytoplasm of live cells and can be detected by any instrument capable of detecting fluorescein [44]. Following treatment, cells were incubated with the diluted DAF-FM diacetate (10μM in serum-free RPMI-1640 medium) for 30 min at 37 °C. DAF-FM diacetate-treated cells were then rinsed twice in a cell staining buffer (420201, BioLegend, San Diego, CA, USA) containing the viability dye 7-aminoactinomycin D (7-AAD, 00-6993-50, eBioscience) and fixed for 20 min at 37 °C in 0.5 mL of a 4% paraformaldehyde-containing fixation buffer (420801, BioLegend, San Diego, CA, USA). Fluorescent activated cell sorting of DAF-FM labeled viable cells was performed on a Beckman Coulter (BC) FC500 flow cytometer (Beckman Coulter, Ottawa, ON, Canada) using Summit software 4.3. Post-acquisition data analysis was performed using FlowJo™ v 10.4.2. The fluorescence emission and excitation maxima for DAF-FM diacetate were in the range of 495–520 nm.

### 2.6. Fluorescent Microscopy

Detection of the fluorescent benzotriazole compounds generated by the chemical interactions between DAF-FM diacetate and intracellular NO was used to determine the effects of tacrolimus on NO production in the HTR-8/SVneo cells. In brief, cells were allowed to adhere to a glass culture cover slip before being treated for 24 h with the conditioned medium containing tacrolimus, L-NAME, and a combination of both, respectively, before undergoing subsequent treatment with the fluorescent dye, DAF-FM. Adherent cells were then fixed in a fixation buffer (420801, BioLegend, San Diego, CA, USA) for 30 min at room temperature and dehydrated with a graded series of ethanols. The cells were then immersed in xylene before being mounted to the slide. Images of cells were captured and analyzed for their fluorescent intensity using Quorum Wave Effects Spinning disc confocal microscope.

The Immunofluorescent detection of PGR and FKBP52 and their colocalization in fixed and rehydrated confluent monolayers of HTR-8/SVneo cells followed an established protocol [5]. Briefly, cells were incubated for 1 h at room temperature in a blocking solution with 5% fetal calf serum in PBS containing 0.05% Tween-20 (PBST). Incubation with anti-PGR- Alexa Fluor^®^ 647 and anti-FKBP52 antibodies (Appendix A) was carried out at 4 °C overnight in a dark humidified chamber. Monolayers of HTR-8/SVneo cells were then rinsed twice in PBS followed by incubation with FITC-conjugated mouse anti-goat antibody for 45 min in a humidified dark chamber at room temperature. Nuclei were counterstained with 4′, 6-diamidino-2-phenylindole (DAPI) (Thermo-Scientific Fisher, Ottawa, ON, Canada). Fluorescence emissions were analyzed using an inverted laser-scanning confocal microscope (Leica model SP2 AOBS). After subtracting non-specific fluorescence obtained in the isotype controls, pixel intensities of yellow fluorescence representing percentages of colocalization of red (PGR) and green (FKBP52) fluorescent signals in overlaid images were quantified by intensity correlation analysis using Image-J analytic software (National Institutes of Health; http://rsb.info.nih.gov/ij) (Accessed on and after 24 April 2021)

### 2.7. Detection of PGR mRNA in HTR8/SVneo Cells

RNA isolation, cDNA synthesis and amplification, and qRT-PCR reaction were performed using CellsDirect™ One-Step qRT-PCR Kit (Catalog number: 11753100, Invitrogen, Burlington, ON, Canada) according to the manufacturer’s instructions. Reverse transcription was carried out for 50 min at 42 °C with SuperScript™ III RT/Platinum™ Taq Mix reverse transcriptase. After an initial 1 min of denaturation step at 94 °C, 35-cycle qtPCRs were carried out on a Bio-Rad’s CFX Connect™ real-time PCR detection system under the following conditions: 1 min denaturation at 94 °C, 1 min annealing at 56 °C, and a 2 min extension at 72 °C. Primers for PCR reactions were as follows; PRB [45]: sense-5′-ACAAGATCTCCACCCAGAGCCCGAGGTTT-3′ and antisense-5′-ACACAATTCATGAGCCGGTCCGGGTGCAAG-3′ (744–1173, 429 bp); PRAB: sense 5′-ACAGAATTCATGACTGAGCTGAAGGCAAAGGGT-3′ and antisense-5′-ACAAGATCTCAAACAGGCACCAAGAGCTGCTGA-3′ (1239–1482, 243 bp); GAPDH [46]: sense- 5′-CAGGGCTGCTTTTAACTCTG-3′ and antisense- 5′-GATGATCTTGAGGCTGTTGTC-3′ (192–723, 532 bp). mRNA copies were calculated using the comparative threshold cycle (ΔΔCT) method with normalization to GAPDH. R values were measured as the negative values of ΔΔCT as exponent of 2 according to the equation: R = R = 2ˆ (−ΔΔCr) where ΔΔCT = ΔCT (Target) − ΔCT (Endogenous Control). GAPDH primers were utilized as positive controls. Negative controls without RNA and without reverse transcriptase were also performed.

### 2.8. Statistical Analyses 

GraphPad Prism (Version 9.4.0) software (GraphPad Software, San Diego, CA, USA) and StatPlus (AnalystSoft.com) (Accessed on and after 24 April 2021) were used for all statistical analyses. Normal distributions were validated using the Kolmogorov–Smirnov method and parameters of normally distributed data are expressed, unless otherwise indicated, as Mean ± S.D. One-way ANOVA followed by Dunn’s multiple comparison test or Mann–Whitney U test or Miller’s procedure were used for pairwise comparisons of independent parameters. The Kruskal–Wallis test was used to analyze mean fluorescence intensity for NO production in the HTR-8/SVneo cells followed by Fisher or Dunn’s post-tests. The *p* value of less than 0.05 was considered statistically significant.

## 3. Results

### 3.1. Low-Dose Tacrolimus Abrogates the Suppressive Effect of L-NAME and Promotes the Human-Derived First-Trimester Extravillous Trophoblast Cell Migration and Invasion In Vitro

Trophoblast migration and invasion are critical to placental development [47]. Therefore, the effects of tacrolimus and L-NAME or mifepristone on the migration and invasion of the EVT were assessed in real-time using the scratch wound protocol and the IncuCyte Zoom Live-cell imager. As shown in Figure 1, unlike treatment with L-NAME or mifepristone which has significantly (*p* < 0.001) delayed wound closure (Figure 1A,B) and suppressed the cell density-dependent migration and invasion of the HTR-8/SVneo cells in vitro (Figure 1B–D), these two metrics were statistically significantly (*p* < 0.001) accelerated in the presence of low-dose tacrolimus (10 ng/mL) irrespective of the presence of L-NAME (Figure 1A–D). Notably, the trophoblast migratory restorative effects of tacrolimus were less pronounced in the presence of mifepristone suggesting the plausibility of PGR dependence in tacrolimus’s mode of action (Figure 1A–D).

### 3.2. Low-Dose Tacrolimus Rescued the Expression of the Active Forms of the Cytosolic Progesterone Receptors and STAT3 in the HTR-8/SVneo Cells Cultured under Nitrosative Stresses In Vitro 

Due to the known importance of the progesterone pathway in pregnancy, we investigated the in vitro effects of tacrolimus on the protein expression and phosphorylation of specific components of this pathway in conditions mimicry of FGR-complicated gestations. Specifically, the active isoform of the PGR-A, phosphorylated PGR-A (pPGR-A) was analysed in the human trophoblast-derived HTR-8/SVneo cells for the different treatment conditions. We hypothesized that tacrolimus has the potential of rescuing the expression and phosphorylation of PGR in conditions imitating those of fetal growth restrictions in vitro. We investigated the expression of PGR in the HTR-8/SVneo cells at the mRNA and protein levels to confirm its expression and assess the effects of tacrolimus in these cells (see Appendix A for PGR mRNA expression in HTR-8/SVneo cells) As shown in Figure 2A–C, treatment with low-dose tacrolimus alone significantly increased (*p* < 0.0001) the expression ratio of pPGR-A/PGR-A in the HTR-8/SVneo cells across the 12–48 h time points. Treatment with L-NAME or mifepristone significantly reduced (*p* < 0.001) the expression of pPGR-A in these cells whereas a combination treatment of low-dose tacrolimus with any of the two inhibitors showed a recovery in the intensity of the bands across all time points (Figure 2A–C).

To further investigate the mode of action of tacrolimus, the levels of phosphorylated STAT3 (pSTAT3-Y705) and unphosphorylated STAT3 were analyzed in the 24-h-treated HTR-8/SVneo cells. STAT3 is an important transcriptional factor involved in the progesterone signaling pathway that is known to regulate trophoblast cell migration and invasion [48,49]. The intensity of pSTAT3-Y705 and STAT3 was relatively constant for the control and low-dose tacrolimus treated cells as seen in Figure 3A and bargraphs in Figure 3B. This outcome was least expected in view of the reported suppressive effects of high-dose tacrolimus (100 nM) on STAT3 phosphorylation in a variety of cellular context [50]. Treatment with L-NAME significantly suppressed (*p* < 0.05) the expression of both the pSTAT3-Y705 and STAT3, whereas combination treatment with L-NAME and low-dose tacrolimus was able to recover the intensity of the bands, similar to that of the untreated control (Figure 3A,B). Moreover, the effect of blocking the actions of progesterone on STAT3 protein expression and activation using mifepristone was also evaluated. As depicted in the representative Western blots In Figure 3C and their densitometric analysis bargraphs in Figure 3D, a minimum of 24 h incubation with mifepristone was sufficient to significantly (*p* < 0.001) downregulate the protein expression of STAT3 and suppresses its activation via phosphorylation at STAT3-Y507 residues as shown in Figure 3C,D, respectively. Interestingly, concurrent with our present in vitro and previous in vivo data on the STAT3 stimulatory effects of low-dose tacrolimus during implantation [5], tacrolimus was found to possess a significant (*p* < 0.001) restorative effect on STAT3 protein expression and phosphorylation in the mifepristone suppressed HTR-8/SVneo cells (Figure 3C and their densitometric analysis in Figure 3D, respectively).

### 3.3. Treatment with Low-Dose Tacrolimus Increased the Expression of FKBP52 and Its Colocalization to the Cytosolic Progesterone Receptors in the HTR-8/SVneo Cells In Vitro

FKBP52 is a tacrolimus (FK506)-binding immunophilin co-chaperone protein and a major component and a downstream target of the cytosolic/nuclear progesterone receptor complex that is critical for gestational success [5,51,52]. It has been suggested that the structural integrity of the mature and active form of the cytosolic PGR for downstream genomic signaling requires the formation of the PGR/FKBP52 chaperone complex [4,51,52]. Therefore, to further discern the mode of action of tacrolimus in promoting the formation of the active form of the cytosolic PGR directly through binding to the FKBP52, we examined the expression of the FKBP52 and its colocalization to the cytosolic PGR in HTR-8/SVneo cells. As shown in Figure 4, monotherapeutic interventions with low-dose tacrolimus significantly induced (*p* < 0.001) the protein expression of the FKBP52 and its colocalization to the cytosolic PGR (Figure 4A–C). Importantly, as a confirmatory finding to validate our hypothesis, we found that the use of low-dose tacrolimus can abrogate the suppressive effect of L-NAME and that of mifepristone on the expression of the FKBP52 and the cytosolic PGR and their colocalization in treated HTR-8/SVneo cells (Figure 4A–C). Therefore, consistent with our current observations from the migration and invasion assays, it appears that the mode of action of tacrolimus in promoting the human-derived first-trimester EVT cell migration and invasion is likely PGR-dependent and that it involves the induction of the active form of the cytosolic PGR and its dynamic interactions with its co-chaperon immunophilin FKBP52. 

### 3.4. Treatment with Low-Dose Tacrolimus Rescued the Production of NO in HTR-8/SVneo Cells in the Presence of L-NAME

Trophoblast production of NO is a recognized contributor of uterine proper spiral artery remodeling during early pregnancy [9]. To examine the effect of low-dose tacrolimus on NO production in HTR-8/SVneo cells, cells treated under the different experimental conditions were collected after 24 h and subjected to flow cytometry analysis and fluorescent staining. The right shifted peak seen in Figure 5A indicates high mean fluorescence intensity for the levels of NO within these cells. Although treatment with tacrolimus alone reduced the level of NO production in the control HTR-8/SVneo cells (Figure 5B,C), it is evident that treatment with low-dose tacrolimus (10 ng/mL) was able to significantly abrogate (*p* < 0.001) the suppressive effect of L-NAME on NO release in HTR-8/SVneo cells as indicated by the higher level of mean fluorescence intensity of NO in these cells (Figure 4A, bargraphs in Figure 4B–D). Therefore, these data suggest that low-dose tacrolimus has a greater influence in allowing for NO production in the presence of a competitive NOS inhibitor in the human-derived first-trimester EVT cells.

### 3.5. Low-Dose Tacrolimus Induced the Phosphorylation of the Stimulatory Domain of eNOS (p-Ser1179) in HTR-8/SVneo Cells Irrespective of the Presence of L-NAME

Due to the important role of eNOS in the synthesis of NO, the level of phosphorylation of both the major stimulatory site, Ser1179 of eNOS, and the inhibitory site, Thr495, were examined in HTR-8/SVneo cells through Western blot analysis. eNOS pSer^1179^, eNOS pThr^495^, and total eNOS were analysed in HTR-8/SVneo cells after undergoing different treatment conditions for 12, 24 and 48 h, respectively. The results obtained with Western blot analysis of eNOS pSer^1179^ is shown in the top panel of Figure 6A. Treatment with low-dose tacrolimus showed a higher intensity of the corresponding bands mainly in the 24–48 h treated cells. Treatment with L-NAME significantly reduced (*p* < 0.01) the expression ratio of eNOS-pSer^1179^/eNOS in these same cells after 24 h of treatment (Figure 6A and bargraphs depicting band intensities in Figure 6B). In contrast, combination treatment with both L-NAME and low-dose tacrolimus showed a recovery in the eNOS-pSer^1179^/eNOS ratio in the 24–48 h treated cells. On the other hand, we were unable to detect statistically significant differences in the relative expression of the inhibitory domain of the eNOS (eNOS-pThr^495^) amongst all treatment conditions (see middle panel of Figure 6A and bargraphs in Figure 6C). Moreover, the suppressive effects of L-NAME on the expression of eNOS and the restoration induced by tacrolimus in the HTR-8/SVneo cells cultured under nitrosative stress is depicted in Figure 6D.

## 4. Discussion

Previous therapeutic interventions have shown the importance of using the immunosuppressive agent tacrolimus in mitigating the severity and incidence of implantation failure, preventing placentation defects, and restoring proper spiral artery remodeling in FGR-complicated gestations [5,20]. However, the mechanism(s) by which tacrolimus is involved in these processes is poorly defined. This study provided cellular and molecular cues evidently showing that tacrolimus has stimulatory effects on the migration and invasion of the human-derived first-trimester extravillous trophoblast cells directly through modulating their NO release and indirectly via the activation of the cytosolic progesterone receptor pathway. 

Our studies on the tacrolimus-mediated promotion of NO synthesis in the HTR-8/SVneo cells cultured under nitrosative stresses have provided further insights into the mechanism(s) by which tacrolimus can support the functional capacities of the human first-trimester EVT cells in FGR-complicated gestations. NO release by trophoblast eNOS is critical to gestational success as it actively regulates embryo development, implantation, trophoblast invasion, assists in maintaining vasodilation and prevents the coagulation of platelets on the trophoblast surface during the invasion process [13,14,15,53]. Importantly, NO synthesis is a shared pathway through which several angiogenic and vasculogenic factors regulating placental angiogenesis and vasculogenesis such as vascular endothelial growth factor (VEGF) and its receptors VEGF-R1 (Flt-1) and VEGF-R2 (fLK-1), transforming growth factor β-1 (TGF β-1), angiopoietin (Ang-) 1 and 2 all exert their effects [54,55]. Currently, there are no data available on the potential regulatory role of tacrolimus on NO synthesis in the human-derived first-trimester EVT cells. Conflicting evidence emerged from previous preclinical and molecular studies regarding the modulatory effects of tacrolimus on NO production and the NOS expression and activity [33,34,35]. Can et al. (2007) showed that the therapeutic use of low–dose tacrolimus (0.01–0.1 μM/L) did not interfere with endothelial NO synthesis in rat aorta and coronary arteries [33]. However, results derived from the work of Tepperman et al. (2011) indicated that tacrolimus has the potential to preserve vasomotor function and maintain vascular homeostasis in a post-transplant murine model [35]. Conversely, Cook et al. (2009) found that a higher experimental concentration of tacrolimus (1 μM/L) decreases vascular NOS activity by reducing agonist-induced intracellular calcium levels and preventing stimulatory changes in eNOS phosphorylation [34]. Our data support the findings of Can et al. (2009) [33] and Tepperman et al. (2011) [35] in that low-dose tacrolimus could either be neutral or has the potential to positively influence NO synthesis through favourably altering eNOS phosphorylation [33,35]. The formation of NO through a two-step oxygenation of L-arginine catalyzed by NOS is contingent upon Ca^2+^/calmodulin interactions and is regulated by several factors including changes in phosphorylation status of the NOS [16]. In the absence of NOS inhibitors, tacrolimus has the potential to reduce NOS functions by binding to FKBP12/12.6 [34] This NOS-suppressive effect of high dose tacrolimus (~25 ng/mL) is believed to be protein kinase C (PKC)-dependent and involves altering intracellular Ca^2+^ release and increased phosphorylation of eNOS at Thr495 [34]. However, in our present use of the NOS inhibitor L-NAME, tacrolimus showed promise in over-riding its suppressive effect on the eNOS stimulatory domain Ser1179 and preserving NO production in HTR-8/SVneo cells cultured under nitrosative stresses. Previous in vitro and in vivo studies suggested that treatment with L-NAME can be used to simulate conditions of NO-deficient hypertension and PE-like gestations with features of FGR [40,56]. L-NAME is a weak competitive inhibitor of eNOS and can only induce nitrosative and oxidative stresses by reducing NO production through enzymatic conversion to its active metabolite and potent eNOS suppressor L-NG-Nitroarginine (L-NNA) [57,58]. This enzymatic conversion of L-NAME into L-NNA by cellular estrases allows the latter to noncovalently bind to the NADPH-binding reductase domain of the eNOS which is regulated by calmodulin [59]. Although it is unclear how the presence of tacrolimus (FK506) rescued the expression of the p-eNOS-Ser1179, data from previous structural and molecular studies indicate that the formation of the FK506-FKBP12 complex may result in a slight increase rather than a decrease in the phosphatase activity of calcineurin in the presence of excess of nitrophenol derivatives such as para-nitrophenolate and its precursor para-nitrophenyl phosphatase which can be induced in the presence of L-NAME [60,61]. This may partially explain the presently observed stimulatory effect of low-dose tacrolimus on eNOS activation and NO release in the L-NAME-treated HTR-8/SVneo cells. This is because calcineurin (Ca^2+^/calmodulin-dependent protein phosphatase) is indirectly involved in stimulating the eNOS enzymatic activity through dephosphorylation at Serine 116 (S116) which promotes eNOS interaction with the c-Src tyrosine kinase and the subsequent eNOS activation through Src-mediated phosphorylation at the eNOS-Y83 residue [62]. Additionally, the Ca^2+^/calmodulin positive regulation of the NADPH-binding reductase domain of eNOS is also a function of the cyclic AMP-dependent protein kinase A (PKA) which can be induced by members of the 14-membered-ring macrolides such as tacrolimus [63]. Furthermore, the restoration of the L-NAME-suppressed eNOS activity by tacrolimus could also be due to its capacity to stimulate the heme-oxygenase (HO) enzymatic system [64] which regulates the oxygenase domain of the NOS [58]. This can potentially protect against the NOS inhibitory effect of L-NAME presently reported in the HTR-8/SVneo cells. However, an indirect effect of tacrolimus on eNOS expression and activation via binding to FKBP12 and associated STAT3 activation could not be excluded at present. Both molecular interactions are known to be associated with altered eNOS expression and activation through, at least in part, the modulation of IL-6 expression in a variety of cellular context [65,66]. The studies by Saura et al. (2006) provided evidence on the STAT3-dependenat IL6-mediated inhibition of the transactivation of the human eNOS promoter in human aortic endothelial cells [65]. Similarly, in a murine model of FKBP12 deficiency, Chiasson et al. (2011) demonstrated that aberrantly elevated IL-6 inhibited eNOS protein expression in a STAT3 dependent mechanism in splenocytes [66]. In our experience, despite a statistical trend (*p* = 0.06) in inducing IL6 protein expression in HTR-8/SVneo cells, a low-dose tacrolimus of 10 ng/mL was not sufficient to significantly induce the expression of this pleotropic cytokine in these EVT cells when cultured under nitrosative stresses in vitro (Appendix A). Conversely, the presently reported solitary inhibitory effects of low-dose tacrolimus on the post-translational modification of eNOS could also be mediated by other eNOS regulatory cytokines paradoxically induced in the HTR-8/SVneo cells, namely TNFα which was found to be significantly stimulated by our monotherapeutic interventions with low-dose tacrolimus (Appendix A). Whereas the suppressive effects of TNFα-on eNOS is believed to be NFΚB-mediated in vascular endothelial cells [67], further research is warranted to explore potential trophoblast-specific immune mechanisms associated with the eNOS modulatory effects of tacrolimus in these cells. Collectively, while this awaits further investigation, it is important to reconcile the fact that the use of low-dose tacrolimus in restoring uterine arterial pulsatility index (UAPI) in the FGR-complicated gestations [20] could be related, at least in part, to its NOS-modulatory activities as concluded in this study and others [35]. This may have significant translational implications considering the altered expression of endothelial NOS with high iNOS and potentially low eNOS activity, and the perturbed L-arginine/NO pathway as well as high levels of NO-metabolites in the cord blood reported in cases of FGR-complicated gestations and in response to chronic hypoxia in women and in animal models [39,68,69,70,71,72,73].

Tacrolimus has also demonstrated the potential to promote the migration and invasion of the human-derived extravillous trophoblasts indirectly by influencing the PGR signaling cascade and prompting the formation of the activated form of the receptor for downstream signaling. The latter mechanism is likely FKBP52-dependent and plausibly involves the nuclear translocation of this tetratricopeptide repeats (TPR)-containing immunophilin in the HTR-8/SVneo cells. While the nuclear translocation of the active form of the cytosolic PGR is contingent upon the structural stability of the PGR/FKBP52 complex and requires the displacement of the FKBP51, another TPR immunophilin with a negative PGR modulatory effect [74,75], the nucleocytoplasmic shuffling of the FKBP52 itself is an independent mechanism and can usually occur in trophoblast cells [76]. This effectively allows dynein, a cytoskeletal motor protein, to bring the entire PGR/FKBP52 complex into the nucleus [76]. This may explain the increased peri-nuclear co-localization of the PGR/FKBP52 presently observed in the tacrolimus-treated HTR-8/SVneo cells indicating an indirect effect of tacrolimus on PGR-mediated influences in the human-derived first-trimester EVT cells. 

In the present study, the L-NAME-suppressed migration and invasion of the HTR-8/SVneo cells were associated with lack of activation and compromised intracellular dynamic interactions of the cytosolic PGR and its co-chaperon FKBP52 in these cells. This action reciprocates that of mifepristone suggesting that the mode of action of L-NAME in suppressing the expression and activation of the cytosolic PGR could be FKBP52-dependent and may involve jeopardizing the structural integrity of the PGR/FKBP52 complex and its nucleocytoplasmic translocation in the HTR-8/SVneo cells. Mifepristone has been shown to have high affinity for progesterone receptors [77,78]. As a potent PGR antagonist, the binding of mifepristone to the receptor interferes with its ability to form stable receptor dimers, thereby inducing a conformational change that results in an inactivation of the receptor such that it is unable to transcribe progesterone-inducible genes [79]. Analogously, while further research is warranted, it may be intuitive to speculate that, besides its competitive NOS suppressive actions, the L-NAME-impeded migration and invasion of the human-derived EVT cells could also be attributed to its reversible in vitro inhibitory effects on the cellular dynamics of the cytosolic PGR in these cells. 

Reminiscent of its in vivo actions in preventing implantation failure and uterine spiral artery remodeling defects in FGR-complicated gestations in mice [5,20], the present use of low-dose tacrolimus significantly induced the expression and phosphorylation of PGR, and to a lesser extent, STAT3-Y705 in the HTR-8/SVneo cells subsequently increasing their migratory and invasive capacities in vitro. The activation of STAT3-Y705 and PGR via phosphorylation is critical to gestational success as it allows for an enhancement of cell motility, consequently increasing the migratory and invasive capability of the trophoblasts [80,81,82]. Despite a plethora of information on the chemical structure, cellular effects and molecular interactions, no data are currently available on the pharmacodynamics of tacrolimus, particularly its effect on the protein expression and phosphorylation of STAT3 and the PGR in the nitrophenyl-stressed human extravillous trophoblast cells. The effect of tacrolimus on STAT3 phosphorylation is controversially described in the literature. Kim et al. (2010) reported a moderate (55%) suppressive effect of low-dose tacrolimus (10 ng/mL) on STAT3-Y705 phosphorylation in isolated human natural killer (NK) cells in vitro [83]. However, a previous study carried out by Liang et al. (1999) indicated that the calcineurin-suppressive effects of tacrolimus (100 µM) resulted in a sustained phosphorylation of STAT3-Y705 in agonist-stimulated vascular smooth muscle cells in vitro [84]. The cellular dynamics of STAT3 activation is highly complex. It involves activation by phosphorylation on the crucial tyrosine residue Y705 which is mainly regulated by members of Janus-activated kinases, particularly JAK1 as a key modulator that triggers STAT3 dimerization and nuclear translocation [85]. The activation can be mediated through Serine (Ser 727) phosphorylation which is commonly regulated by Akt, protein kinase C, mitogen-activated protein kinases, and CDK5 [86,87,88], as well as the revocable acetylation by histone acetyltransferase on an individual lysine residue (STAT3-Lys 685) [89]. It also involves deactivation through dephosphorylation by tyrosine and serine phosphatases including calcineurin [84]. In our experience, the tacrolimus-induced expression and phosphorylation of STAT3 was associated with PGR-mediated activation of Akt in the HTR-8/SVneo cells (Appendix A). However, considering the complexity of the cellular dynamics of STAT3, particularly the strict requirement for dephosphorylation for transcriptional inactivation [90], and the confounding reports on the role of tacrolimus in modulating STAT3 activation via phosphorylation and acetylation [83,84] further research is warranted to discern the probable mechanism(s) of the presently reported tacrolimus-mediated activation of STAT3-Y705 in HTR-8/SVneo cells. Particularly, the effects of low-dose tacrolimus on STAT3 deacetylation and the expression of SOC (suppressors of cytokines) proteins in the human-derived EVT cells is warranted.

Lastly, in their recent in vivo study on the L-NAME-induced PE-like conditions in rats, Zhang et al. (2018) reported a reduction in the protein expression of placental STAT3 and its active form p-STAT3-Y705 [91]. On the other hand, Park et al. (2018) showed that the use of L-NAME (100 μM/mL) did not significantly alter the protein expression of the PGR-B in human placenta-derived cytotrophoblast-like BeWo cells treated for 24 h under hypoxic conditions in vitro [92]. Although our data align with the findings of Zhang et al. (2018) regarding the lack of STAT3 activation in the L-NAME-treated placentae [91], it is likely that the cellular and molecular discrepancies between the HTR-8/SVneo cells and the BeWo cells particularly their downstream PGR-mediated responses [93,94], may have accounted for the differences in the L-NAME-mediated alterations in the PGR expression reported in the present study and that of Park et al. (2018) [92].

## 5. Conclusions

Data obtained in the present study suggest an immune-independent mode of action of tacrolimus in preventing pregnancy failure and fetal growth restriction in complicated gestations by facilitating the remodeling of the uterine spiral arteries, at least in part, through the promotion of the first trimester extravillous trophoblast cell migration and invasion and nitric oxide production via NOS- and progesterone receptor-mediated influences. Our current in vitro evidence of the positive influence of low-dose tacrolimus in promoting the migration and invasion of the human-derived first-trimester EVT cells supports our previous in vivo findings in a murine model of FGR-complicated gestation [5] highlighting the promising potential of tacrolimus in potentiating the enrichment of PGR signaling indispensable for the EVT-assisted remodeling of the uterine spiral arteries during the hemochorial placentation in mice and women. Moreover, we have provided molecular cues supporting the hypothesis that tacrolimus can rescue the production of nitric oxide through stimulating the phosphorylation of the active domain of nitric oxide synthase in the human-derived extravillous trophoblast cells challenged with nitrosative stresses in vitro. Additionally, our data suggest that the experimental use of L-NAME in instigating gestational adversities mimicry of the FGR-complicated pregnancies could additionally be attributed to its suppressive effects on the expression and phosphorylation of the cytosolic progesterone receptors and their dynamic interactions with the co-chaperone immunophilin FKBP52. Furthermore, our results demonstrated a potential for tacrolimus to overcome the suppressive effects of L-NAME on the migration and invasion of the human-derived first-trimester EVT cells plausibly through the activation of STAT3-Y705 and the induction of the phosphorylation of the active domain of the endothelial NOS (p-eNOS-Ser^1197^). Lastly, in addition to previous reports on the cytoprotective and stimulatory effects of tacrolimus on cell migration through increasing the production of stem cell factor and MMP-9 activity in a variety of cellular context [95], the present study has provided further molecular clues implicating the modulation of NO production and the potentiation of the PGR-FKBP52 nucleocytoplasmic shuffling as alternative mode of actions of tacrolimus in promoting migration and invasion of the human-derived first-trimester extravillous trophoblast cells in vitro. Collectively, results obtained in the present study show great promise for the use of low-dose tacrolimus in positively influencing the migratory and invasive functions of the human first-trimester extravillous trophoblast cells conducive to proper uterine spiral arterial remodeling and placental formation in the FGR- complicated gestations.

## Figures and Tables

**Figure 1 ijms-23-08426-f001:**
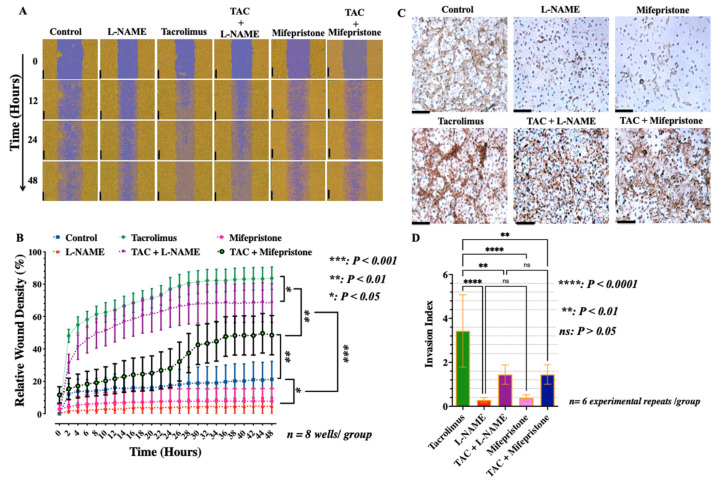
Tacrolimus promotes migration and invasion of the human-derived first-trimester extravillous trophoblasts in vitro. HTR-8/SVneo cells were grown to confluence in complete and unconditioned medium (DMSO-conditioned) and were plated (1 × 10^5^ cells per well) onto 24-well plate prior to treatment with conditioned medium containing tacrolimus (10 ng/mL), L-NAME (50 mM/mL), or mifepristone (50 nM/mL) and in combination formulations of tacrolimus and the inhibitor. Scratch wounds (purple area in (**A**)) were made on confluent monolayers using a 96-pin WoundMaker and cells were incubated at 37 °C in 5% CO_2_ and imaged by IncuCyte™ Zoom live cell scanner every 2 h for 48 h for quantitative analyses. (**A**) Representative IncuCyte™ images of wound closure over time. The dark-yellow region denotes the scratch wound mask (i.e., wound area covered by advancing cells) created immediately following wound creation and over time (0–48 h) as the HTR-8/SVneo cells migrated into the wound region. (**B**) The time course of the HTR-8/SVneo cells migration in conditioned medium. Compared to the control, L-NAME- and mifepristone-treated cells, low-dose tacrolimus (10 ng/mL) significantly reduced the amount of time for trophoblast cells to reach 90% confluence recorded at 24 h after wound creation in treated monolayer of cells. (**C**) Representative photomicrographs of invaded cells in transwell PET filter 24 h after treatment with tacrolimus, L-NAME or mifepristone, and in combination formulations. (**D**) Bargraphs representation of the Invasion Index depicting the significantly increased invasion rate of the HTR-8/SVneo cells (*p* < 0.001 when compared with respective controls per time points) upon tacrolimus treatment in the presence or absence of L-NAME or mifepristone (*n* = 6/group). Data indicate mean ± standard deviation (SD). Scale bars in A and C = 100 μm.

**Figure 2 ijms-23-08426-f002:**
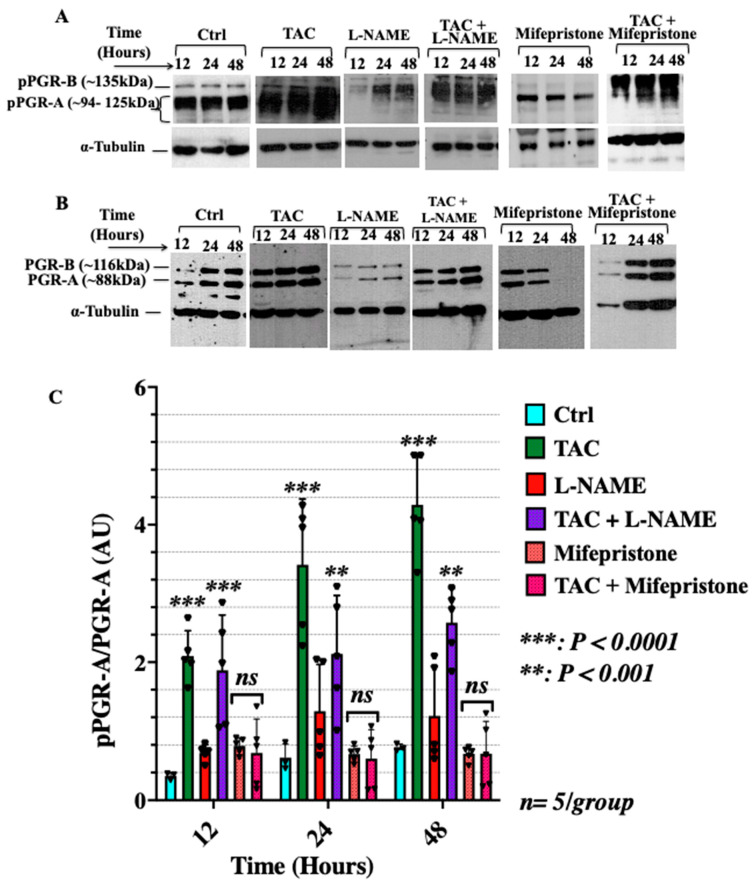
Tacrolimus abrogates the suppressive effects of L-NAME and mifepristone, inducing the expression and activation of PGR in HTR-8/SVneo cells. (**A**,**B**) Representative Western blots of pPGR-A and p-BGR-B (**A**) and PGR-A and PGR-B (**B**) in HTR-8/SVneo cells treated, respectively, with low-dose tacrolimus (10 ng/mL), L-NAME or mifepristone, and in combination formulations for 12, 24 and 48 h. (**C**) Histogram depicting the relative expression ratio of pPGR-A/PGR-A shown in (**A**,**B**).

**Figure 3 ijms-23-08426-f003:**
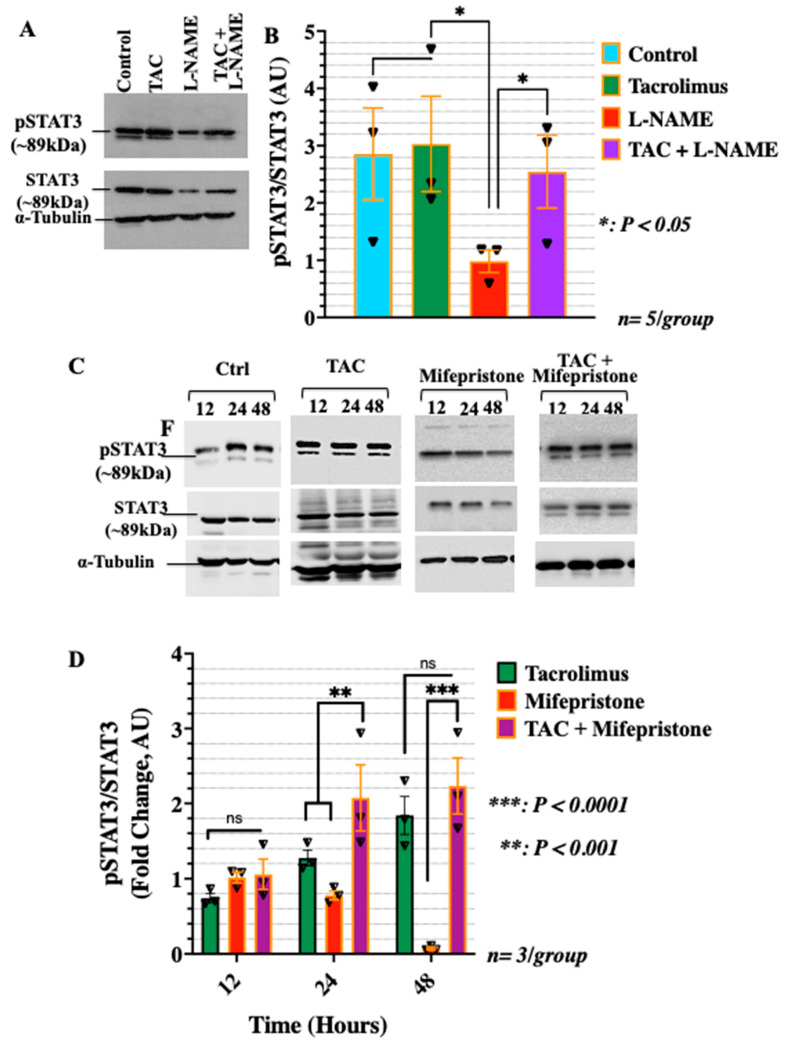
Tacrolimus abrogates the suppressive effects of L-NAME and mifepristone without altering the expression and activation of STAT3 in HTR-8/SVneo cells. (**A**,**B**): Representative Western blot (**A**) and densitometric analysis (**B**) of pSTAT3-Y705 and STAT3 in the HTR-8/SVneo cells treated, respectively, with tacrolimus, L-NAME, and a combination of both after 24 h and the untreated control. It is evident that tacrolimus has the potential to counteract the suppressive effect of L-NAME on the expression and activation of the PGR and STAT3 in HTR-8/SVneo cells cultured under nitrosative stresses in vitro. Similarly, tacrolimus was able to override the inhibitory actions of mifepristone on STAT3 expression and phosphorylation at STAT3-Y705 in HTR-8/SVneo cells (**C**,**D**). Band intensity was generated using the ImageJ software. Data represent the mean intensity with error bars representing the S.D. TAC: tacrolimus.

**Figure 4 ijms-23-08426-f004:**
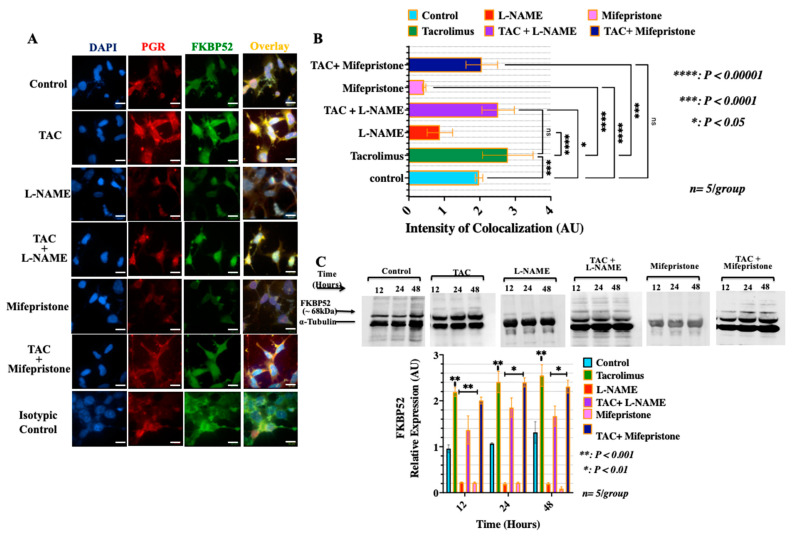
Effect of low-dose tacrolimus, L-NAME or mifepristone and their combination formulations on the levels of expression and cellular localization of FKBP52 and PGR in HTR-8/SVneo cells. (**A**) Representative confocal fluorescent microscope images of monolayers of confluent HTR-8/SVneo cells treated for 24 h with low-dose tacrolimus (10 ng/mL), L-NAME or a combination of both. Cells were stained with antibodies to PGR (red channel) and FKBP52 (green channel) and cell images were captured and analyzed for their fluorescent intensity using Quorum Wave Effects Spinning disc confocal microscope. Nuclei were counterstained with DAPI (blue channel) and the colocalization of PGR and FKBP52 appears as yellow on the overlay images. (**B**) Histogram depicting differences in the intensity of colocalization of PGR and FKBP52 (yellow channel) showing significant effect of low-dose tacrolimus on the expression of the PGR-FKBP52 complex in treated HTR-8/SVneo cells. (**C**) Representative WB images and associated histograms of the expression levels of FKBP52 protein in the tacrolimus- and L-NAME-treated HTR-8/SVneo cells. Note the statistically significant effects (*p* < 0.001) of tacrolimus in antagonizing the suppressive actions of L-NAME and those of mifepristone on the protein expression of FKBP52 in the HTR-8/SVneo cells. Data in (**B**,**C**) are represented as the mean with error bars representing S.D. Scale bars in A = 15 μm. Mifepristone was used to determine the mode of action of tacrolimus in potentiating the PGR signaling in HTR-8/SVneo cells.

**Figure 5 ijms-23-08426-f005:**
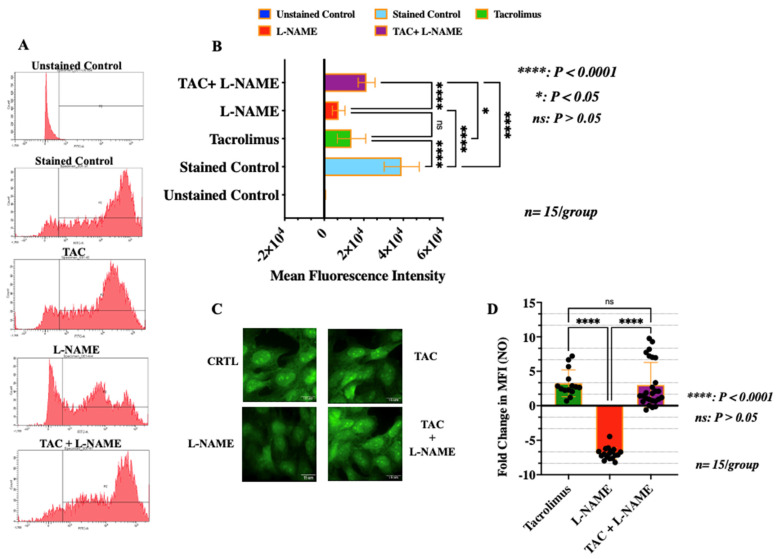
Effect of tacrolimus, L-NAME and combination treatment on NO production in HTR-8/SVneo cells. (**A**) Representative flow-cytometry histogram analysis of the NO levels detected in the unstained and stained control HTR-8/SVneo cells and those after 24 h of treatment with low-dose tacrolimus, L-NAME, and a combination of both. (**B**) Histogram depicting the median fluorescent intensity of the benzotriazole compounds generated by the chemical interactions between DAF-FM diacetate and intracellular NO in the untreated and treated HTR-8/SVneo cells. (**C**,**D**) Fluorescence microscopic detection (**B**) and quantification (**D**) of NO in the control, and in the cells treated, respectively, with tacrolimus, L-NAME and a combination of both using Diamino Fluorescein diacetate (DAF-FM). Data in B represent the median fluorescence intensity, with error bars representing S.D. Scale bars in C = 15 μm.

**Figure 6 ijms-23-08426-f006:**
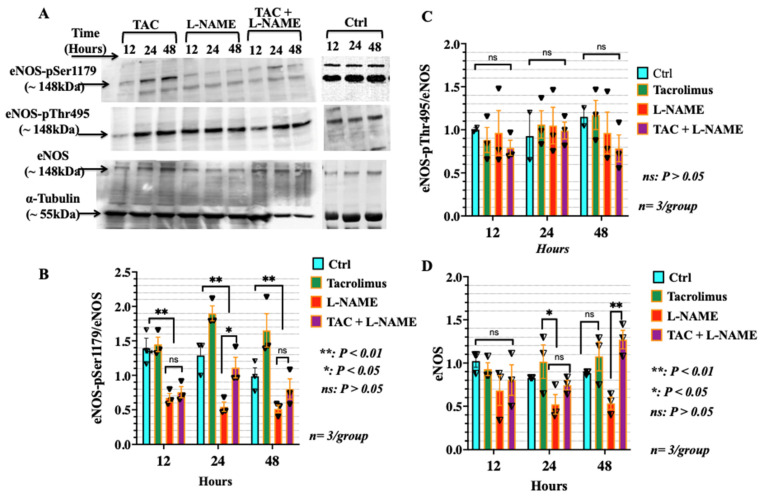
Low-dose tacrolimus stimulates the expression of the eNOS and rescues the phosphorylation of its active domain eNOS p-Ser^1179^ in the L-NAME-challenged HTR-8/SVneo cells. (**A**) Representative WB of protein levels of eNOS p-Ser^1179^ and eNOS p-Thr^495^ and that of the eNOS in the HTR-8/SVneo cells after 12, 24 and 48 h of treatment with low-dose tacrolimus, L-NAME or a combination of both. (**B**,**C**) Histograms showing the effect of tacrolimus in rescuing the phosphorylation of the eNOS p-Ser^1179^ in the L-NAME-treated cells after 24 h of adding tacrolimus into the conditioned medium (**B**). Notably, no statistically significant differences (*p* > 0.05) were observed between tacrolimus and L-NAME on the relative expression levels of the eNOS p-Thr^495^ (**C**). (**D**) Representative Wb showing the suppressive effects of L-NAME on the expression of eNOS and the restoration induced by tacrolimus in the HTR-8/SVneo cells cultured under nitrosative stress. Band intensity was generated using the ImageJ software. Data presented in B and C are the mean intensity with error bars representing the S.D. Ctrl: control, TAC: low-dose tacrolimus, T + L: tacrolimus + L-NAME.

## Data Availability

Due to institutional guidelines on the sharing of research information, the data presented in this study are available on request from the corresponding author.

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
