# Peer review of "Low-Dose Tacrolimus Promotes the Migration and Invasion and Nitric Oxide Production in the Human-Derived First Trimester Extravillous Trophoblast Cells In Vitro"

_ijms, 2022, doi:10.3390/ijms23158426_

Round 1

Reviewer 1 Report

This article showed that low-dose tacrolimus stimulates the migration and invasion of the human-derived extravillous trophoblast (EVT) cells significantly, and the authors suggested that these results obtained from directly through modulating their NO release and indirectly through the activation of the cytosolic progesterone receptor pathway.

The authors have previously shown that the use of tacrolimus restores uterine spiral artery remodeling in a murine model, this time the authors investigate the mechanism of action of tacrolimus on EVT cells in the point of molecular pathway and NO production in-vitro.

In fact, there are no study investigated about the potential regulatory role of tacrolimus on NO synthesis in the human-derived EVT cells, the results from this study are enough interesting. But, several points should be addressed by the authors.

  1. The authors considered the stimulatory effect of tacrolimus on the migration and invasion of the human-derived EVT cells were explained directly through modulating their NO release and indirectly through the activation of the progesterone receptor pathway. But, in functional assay, they use only the NOS inhibitor L-NAME, experiments blocking progesterone receptor were lacking.

  1. In discussion, the authors told that low-dose tacrolimus significantly induced the expression and phosphorylation of STAT3-Y705 in the HTR-8/SVneo cells. However, the difference in the expression of pSTAT3 were not significant between control and with tacrolimus in figure 2 D and E. Indeed, they mentioned this point, it is better to explain the reason.  

  1. In Figure 4, Under tacrolimus treatment, NO synthesis decreased when compared with stained control. Some reason should be addressed.
  2. Scale is necessary in Figure 1A and C.

  1. The word Error bar should be change into scale bar in the legend of figure 4C.

Reviewer 2 Report

Albaghdadi et al., present a paper on the direct effects of tacrolimus on HTR8/SVneo cells functionality and underpin this effect via increase in local NO production. Although interesting study, there are some concerns of mine that I would like to rise below:

Introduction:

  • In the beginning of the introduction (lines 28-44) the authors concentrate on the effect of tacrolimus on spiral artery remodeling, and it feels like they would depict this role of tacrolimus in the paper as well. Because no experiments were done in vitro to depict angiogenesis (e.g. tube formation assay), I would suggest to explain the placentation process more in the introduction (e.g. depicting trophoblast migration, proliferation and invasion).

Methods:

  • Line 117, authors mention only three groups, although in the rest of the figures we see another group mentioned control. Please mention this as well in the methods part.
  • L-Name and Tacrolimus are dissolved in DMSO, so the authors need to present another control group where they use the same amount of DMSO as present in the other groups. Also, throughout the paper some figures show control group, but others don’t. What this control group received is rather unknown!
  • It is not mentioned the passage number, nor whether the cells were regularly tested for mycoplasma.
  • Section 2.5 I don’t understand why the samples for fluorescent staining were dehydrated instead of mounting them with an aqueous mounting medium. Dehydration affects the fluorescence signal.

Results:

  • Please check figure 1 for inconsistences e.g. figure 1A: It’s not mentioned how big is the scale bar, and in figure 1C the scale bar is missing. Then figure 1B as it is now is really misinforming about the statistical difference between the groups. Please place the statistical differences on the side of the graph and clearly represent among which groups the statistical difference is present.
  • The progesterone receptor expression in HTR8/SVneo is still controversial. Terrence et al (PMID: 23813454) reported that PGR-A and -B cannot be detected in HTR-8/SVneo cells, but the membrane-associated receptor is present PGRMC1. Here the authors report that PGR-A and -B are present and even upregulated in HTR-8/SVneo cells treated with tacrolimus, however the control group are missing to really confirm this. Are there available data from the gene expression of these receptors as well? Is there a possibility that the tacrolimus acts via PGRMC1?
  • Is rather unclear why the authors choose to check the levels of STAT3 and pSTAT3 only and not the rest of the progesterone signaling pathway, such as MAPK or AP-1.
  • Figure 4C: could you quantify the DAF-FM fluorescence intensity among the groups?
  • L-Name is known inhibitor of eNOS, however in figure 5A, we only see downregulation of eNOS-pSer, but not in total eNOS. Can the authors comment on this?
  • Authors use HTR-8/SVneo cells as a trophoblast model, however the paper of Abou-Kheir et al. (PMID: 28161053), shows that these cells show a heterogenous population of stromal and trophoblast cells. The authors should comment on this and observe whether the tacrolimus effect is really trophoblast specific.
  • Tacrolimus as an immunosuppressant can also inhibit cytokine production and several cytokines including TNFa, IL-6 and IL-2 have been reported to regulate NOS production. Did the authors check for cytokine secretion in the supernatant from HTR-8/SVneo cells upon treatment with tacrolimus? Would be nice to see whether the low dose of tacrolimus does not perturb the cytokine milieu in HTR-8/SVneo cells.   

Discussion:

                -Please consider the last paragraph (lines 566-598) to be separated in a separate subsection Conclusion.

Minor comments:

  • Line 119: change mMol to mM.
  • Line 141 and 146: why is CO2 in between brackets?
  • Figure 3C: L-NAME is misspelled in the graph legend.
  • Figure 4A: Please order the histograms in the same way as the graph bars. Meaning after the control, place the Tacrolimus group and then the rest. In that way, you preserve the order in all figures.
  • Line 372: delete however.
  • Line 388: change Error to Scale.
  • Space between 10ng/ml. Please revise throughout the text.
  • Please be consistent with the naming of HTR-8/SVneo cells. Sometimes dash is used and sometimes not. Check this throughout the text.
  • Also, sometimes mL is written with capital L and sometimes not. Please revise throughout the text.
  • Line 606 doesn’t show the author contributions. Please revise.
  • Reference 68 is non-existent.

Reviewer 3 Report

This study aimed to determine the effect of low-dose tacrolimus on trophoblast invasion and migration in vitro. Their results showed that low-dose tacrolimus stimulate the migration and invasion of HTR8/SVneo, by inducing PR expression and activation, and modulating NO production. This set of data are potentially contribute to understand the therapeutic mechanism of tacrolimus in clinical application. However, some major concerns in this study can not be ignored.

The major concern is regarding the cell model in this study. The authors only used HTR8/SVneo to perform the cell treatments, these in vitro results are lack of novelty, and not sufficient to support their conclusions.

The regulatory network among PR, eNOS, STAT3 needs to be further validated.

Figure 3, the localization of PGR was positive both in nucleus and cytoplasm. 

Figure 4. the IF intensity looks similar among control, TAC, and TAC+L-NAME groups.

Round 2

Reviewer 2 Report

No further comments for the authors.